Genetic variation for tolerance to pre-harvest sprouting in mungbean (Vigna radiata) genotypes

Gupta Soma 1
http://orcid.org/0000-0002-9021-2650 Aski Muraleedhar 1
http://orcid.org/0000-0002-7023-2638 Mishra Gyan Prakash 1 2 gyan.gene@gmail.com
Yadav Prachi S. 1
http://orcid.org/0000-0002-9069-1583 Tripathi Kuldeep 3
Lal Sandeep Kumar 2
Jain Simran 1
Nair Ramakrishnan Madhavan 4
Dikshit Harsh Kumar 1 harshgeneticsiari@gmail.com
1 Division of Genetics, ICAR-Indian Agricultural Research Institute , New Delhi, Delhi , India
2 Division of Seed Science and Technology, ICAR-Indian Agricultural Research Institute , New Delhi, Delhi , India
3 Division of Germplasm Evaluation, ICAR-National Bureau of Plant Genetic Resources , New Delhi, Delhi , India
4 South Asia/Central Asia, World Vegetable Center , Patancheru, Hyderabad, Telangana , India
Kumar Sushil
Electronic publication date: 2024 Jul 23
Publication date: 2024
Volume: 12
Electronic Location ID: e17609
Received 2024 Feb 14; Accepted 2024 May 30
Copyright: © 2024 Gupta et al.
Copyright year: 2024
Copyright holder: Gupta et al.
License: This is an open access article distributed under the terms of the Creative Commons Attribution License, which permits unrestricted use, distribution, reproduction and adaptation in any medium and for any purpose provided that it is properly attributed. For attribution, the original author(s), title, publication source (PeerJ) and either DOI or URL of the article must be cited.
License URL: https://creativecommons.org/licenses/by/4.0/

Keywords: Mungbean, Pre-harvest sprouting, Genetic variability, Donors, Correlation, Principal component analysis

Funding: ICAR-Indian Agricultural Research Institute New Delhi and funding was provided by the Australian Centre for International Agricultural Research (ACIAR) for the International Mungbean Improvement Network project CROP/2019/144 The research work has been supported by the ICAR-Indian Agricultural Research Institute, New Delhi and funding was provided by the Australian Centre for International Agricultural Research (ACIAR) for the International Mungbean Improvement Network project (grant number CROP/2019/144). The funders had no role in study design, data collection and analysis, decision to publish, or preparation of the manuscript.

==============================
Pre-harvest sprouting (PHS) is one of the important abiotic stresses in mungbean which significantly reduces yield and quality of the produce. This study was conducted to evaluate the genetic variability for tolerance to pre-harvest sprouting in diverse mungbean genotypes while simultaneously deciphering the association of yield contributing traits with PHS. Eighty-three diverse mungbean genotypes (23 released varieties, 23 advanced breeding lines and 37 exotic germplasm lines) were investigated for tolerance to PHS, water imbibition capacities by pods, pod and seed physical traits. Wide variation in PHS was recorded which ranged between 17.8% to 81% (mean value 54.34%). Germplasm lines exhibited higher tolerance to PHS than the high-yielding released varieties. Correlation analysis revealed PHS to be positively associated with water imbibition capacity by pods (r = 0.21) and germinated pod % (r = 0.78). Pod length (r = −0.13) and seeds per pod (r = −0.13) were negatively influencing PHS. Positive associations between PHS and water imbibition capacity by pods, germinated pod % and 100-seed weight was further confirmed by multivariate analysis. Small-seeded genotypes having 100-seed weight <3 g exhibited higher tolerance to PHS compared to bold-seeded genotypes having 100-seed weight more than 3.5 g. Fresh seed germination among the selected PHS tolerant and susceptible genotypes ranged from 42% (M 204) to 98% (Pusa 1131). A positive association (r = 0.79) was recorded between fresh seed germination and PHS. Genotypes M 1255, M 145, M 422, M 1421 identified as potential genetic donors against PHS could be utilized in mungbean breeding programs.

Introduction

Mungbean (Vigna radiata (L.) R. Wilczek) is a self-pollinated leguminous crop having a relatively small genome size of 493–579 megabase (Mb) pairs (Arumuganathan & Earle, 1991; Kang et al., 2014; Liu et al., 2016). Mungbean is short-duration grain legume well-known for its excellent nutritional value. Mungbean grains are rich in protein, carbohydrates, amino acids, dietary fibers, minerals and vitamins (Hou et al., 2019; Somta et al., 2022). Shorter maturity duration, nitrogen fixing ability, tolerance to heat, drought and wider adaptability to varying climatic regimes makes mungbean a good fit for various cereal-based cropping systems (Mishra et al., 2022). Mungbean is a widely cultivated legume crop in South, East, South-East Asia and now its area is expanding in Australia, United States and Africa. Globally mungbean occupies 7.5–8.0 million ha of production area, predominantly (80–90%) confined to Asia (Nair & Schreinemachers, 2020; Anonymous, 2021). India is the largest producer and consumer with about 5.5 million ha area under mungbean cultivation yielding 3.17 million tons of total production (ICAR-IIPR, 2022).

Pre-harvest sprouting (PHS) is a global threat affecting staple crops including wheat, rice and legumes. PHS involves in situ germination of seeds prior to harvest, triggered when seed maturation coincides with frequent rainfall and high temperatures. PHS is an induced abiotic stress and serious economic concern in mungbean production adversely affecting yield, end-use quality of produce and seed viability. While domesticating mungbean, rapid and uniformly germinating genotypes had been preferably selected over other which has led the crop become less dormant than their wild progenitors.

PHS is a complex trait associated with seed dormancy (Park et al., 2021; Dhariwal et al., 2021). Seed dormancy is attributed to morphological, physiological, morpho-physiological, physical and combinational of physical and physiological properties (Baskin & Baskin, 2004). Physiological seed dormancy is the most prevalent dormancy class resulting due to physiological immaturity/incompetency of the embryo to germinate (Finch-Savage & Leubner-Metzger, 2006). This class of seed dormancy is controlled by environmental and endogenous factors. Phytohormones, gibberellic acid and abscisic acid are key regulators of seed dormancy and PHS. Gibberellic acid stimulates seed germination whereas abscisic acid has inhibitory role (Tuttle et al., 2015; Dallinger et al., 2023; Ye et al., 2015). Physical seed dormancy, also called hard-seededness is caused by impregnation of phenolics and suberin in the palisade layer of seed coat making it impermeable to water molecules (Smykal et al., 2014). Hard-seededness imposed water-impermeability restricts water imbibition and thus delays seed germination and prevents PHS. Hard-seededness, on the other hand, interferes with rapid seedling emergence and prolongs cooking time which makes it an undesirable trait at consumers’ end. Akin to other wild legume species, wild mungbean exhibits strong seed dormancy controlled by a single dominant gene with or without modifying genes (Singh, Sharma & Dwivedi, 1983). Seed dormancy in wild mungbean has been reported as physical (Laosatit et al., 2022) and combinational (both physical and physiological) in nature (Somta et al., 2022). Laosatit et al. (2022) demonstrated the cuticle layer imparting physical dormancy in the wild mungbean. Low levels of seed dormancy lead to PHS whereas high levels result in delayed seed germination and non-uniform seedling establishment in the field. Thus, optimum level of seed dormancy/dormancy at harvest is desirable in any crop including mungbean.

Mungbean germplasm have been assessed for identifying PHS tolerant genotypes Rao, Rao & Rao (2007). Ahmad, Khulbe & Roy (2014) evaluated 112 diverse mungbean genotypes for PHS tolerance and studied its association with various morphological characters, their direct and indirect effects on PHS. Numerous quantitative trait loci (QTLs) associated with seed dormancy/tolerance to PHS have been identified and mapped in different crops such as rice (Lee et al., 2023; Cheon et al., 2020), wheat (Shawai et al., 2023; Li et al., 2021; Dhariwal et al., 2021), barley (Hickey et al., 2012; Nakamura et al., 2016) and cucumber (Cao et al., 2021). Several QTL/QTNs associated with tolerance to PHS have been identified and developed markers have been validated in wheat (Zhu et al., 2019; Jiang et al., 2022; Kumar et al., 2023). In mungbean, one major QTL, HsA, has been reported explaining 23.2% of the dormancy variation (Humphry et al., 2005). Isemura et al. (2012) reported seed dormancy in mungbean to be conditioned by two major and two minor QTLs, of which QTL Sdwa5.1.1+ (on LG1) explained 33.7% of the seed dormancy variation (Isemura et al., 2012). Laosatit et al. (2022) fine mapped the QTL Sdwa5.1.1+ to 3.298-Kb region which contained gene LOC106767068, designated as VrKNAT7-1. They demonstrated class II KNOTTED1-LIKE HOMEOBOX (KNOX II) gene, KNAX7-1 to be associated with physical dormancy in wild mungbean.

In Northern and Eastern India, mungbean is primarily cultivated during rainy season (mid July-mid October), wherein sporadic rain at the time of maturity prompts PHS. In mungbean, PHS has been reported to reduce yield by 60–70% (Durga & Kumar, 1997). Despite of huge losses caused by PHS in mungbean, this is yet a relatively less investigated abiotic stress. Therefore, the present study was undertaken with the following objectives (i) to assess the genetic variation for tolerance to PHS in mungbean, (ii) identification of potential genetic source/donor(s) for tolerance to PHS in mungbean, (iii) to understand the associations of different pod and seed traits with tolerance to PHS.

Materials and Methods

Experimental site and planting material

Eighty-three mungbean genotypes (23 released varieties, 23 advanced breeding lines and 37 exotic germplasm lines) were evaluated for tolerance to PHS, pod and seed physical traits (Table S1). Mungbean genotypes were raised at the experimental farm, ICAR-Indian Agricultural Research Institute (ICAR-IARI), New Delhi (28°40′44.68″N, 77°4′10.95″E) during July 2022. Delhi is located at Trans Ganga Plain of India and is situated at 218 masl receiving an average of 886 mm of annual rainfall. The soil at experimental site is sandy loam in texture and the climate is overlap between humid subtropical and semi-arid. Genotypes were planted in a single row (4 m length) having spacing of 10 cm between the plants and 30 cm between the rows. Recommended package of practices was followed to raise healthy crop. Pods were harvested at maturity. Healthy intact pods with no rupture/pod wall damage were stored at 4 °C and used for conducting the experiment.

Water imbibition capacity by pods (WICP)

Water imbibed by pods was recorded for fifteen pods in triplicate. Dry pods were weighed and placed in moist germination paper inside growth chamber maintained at 25 °C for 24 h. The pods were taken out after incubation of 24 h, blotted dry, weighed and placed back immediately for evaluation of percent pod germination and percent seed germination within pod (PHS). The volume of water imbibed by pods was derived by the weight gained by the moist pods after 24 h of incubation and expressed as percentage.

Germinated pods (GP %) and seeds germinated within pod (PHS %)

Matured intact pods with their peduncles attached, were evaluated to determine percent seed germination within pods (PHS) and percent pod germination after 4 days of incubation. Fifteen pods of each genotype (number of replications-3) were tested for germination following between paper method of seed germination test to evaluate GP and PHS. The temperature in growth chamber was maintained at 25 °C with relative humidity of 80% to mimic field condition conducive to pre-harvest sprouting in mungbean. For GP (%), number of germinated pods, out of fifteen, were counted and expressed in percentage. For PHS (%), total number of seeds germinated within each pod was recorded from each replication. Percent seed germination within pod was calculated.

Pod and seed physical traits

Observations on pod traits (pod length (PL), pod diameter (PD) and number of seeds per pod (SPP)) and seed traits (seed length (SL) and seed width (SW)) were recorded on fifteen pods and ten seeds of each genotype, respectively in three replicates using a digital vernier caliper. Number of seeds per pod was recorded on fifteen pods of each genotype in triplicate. The 100 seed weight (HSW) was recorded using an electronic weighing balance.

Fresh seed germination (FSG)

Mungbean genotypes were re-sown at the experimental farm, ICAR-IARI, New Delhi during July 2023. Pods were harvested at maturity and threshed manually. Freshly harvested seeds of the selected PHS tolerant, i.e., PHS <30% (n = 10) and susceptible, i.e., PHS >75% (n = 9) lines were immediately brought to the laboratory to test for FSG. For each genotype, 50 seeds in triplicate, were placed in petri plates lined with two pre-wetted filter papers. The seeds were incubated at 25 °C in dark condition for 4 days. The number of seeds germinated (radicle length of approximately 2 mm) were counted after 4 days of incubation and expressed as percent germination.

Statistical analysis

Descriptive statistics, one-way analysis of variance (ANOVA), correlation, circular dendrogram and principal component analysis (PCA) were carried out using PAST version 4.03 and Minitab statistical software. Linear regression was drawn using the equation

yi = a + bxi

where, yi is the dependent variable, xi is the independent variable, a is the intercept, b is the regression coefficient of the variable.

Results

Genotypic variation for seed and pod physical traits

The studied panel of genotypes exhibited wide variations for seed and pod physical traits (Fig. 1, Table 1). The pod length and diameter varied from 6.13 to 8.72 cm and 2.9 to 4.8 mm, with an average of 7.2 cm and 3.87 mm, respectively. Significant variation was observed for number of seeds per pod ranging from 7.61 to 11.06 having an average of 9.29. Similarly, seed length and width varied from 3.1 to 5.1 mm and 2.3 to 3.7 mm, with an average of 4.1 and 3 mm, respectively. Among the studied genotypes, 100-seed weight ranged from 2.03 to 4.59 g, with an average value of 2.9 g. Genotypes M 1255 (germplasm line), HUM 2 and Pusa Baisakhi (released varieties) exhibited higher pod and seed traits. Among seed and pod traits, the highest genotypic variation was observed for hundred seed weight (combined CV = 17.28%), followed by pod diameter (combined CV = 9.57%), seed length (combined CV = 9.55%), seed width (combined CV = 8.89%), pod length (combined CV = 7.9%) and seeds per pod (combined CV = 7.53%) (Table 1).

Figure 1 Frequency distribution of studied traits viz. water imbibition capacity by pods (WICP), germinated pod % (GP), pre-harvest sprouting (PHS), pod length (PL), pod diameter (PD), number of seeds per pod (SPP), seed length (SL), seed width (SW), 100-seed weight (HSW) among 83 mungbean genotypes.

Table 1 Descriptive statistics of pre-harvest sprouting, various seed and pod traits in mungbean genotypes.

Variable	Type	Mean	Min	Max	SE	SD	CV	
WICP	RV	48.12	38.28	59.50	0.99	4.75	9.87	
ABL	47.66	38.16	64.64	1.29	6.19	12.99	
GL	45.46	32.59	68.73	1.18	7.20	15.83	
Combined	46.81	32.59	68.73	0.69	6.37	13.61	
GP	RV	76.51	55.55	91.11	1.98	9.54	12.47	
ABL	81.05	44.44	93.33	2.39	11.50	14.19	
GL	77.95	31.10	95.55	2.38	14.50	18.60	
Combined	78.41	31.10	95.55	1.36	12.46	15.89	
PHS	RV	53.85	27.47	80.56	2.88	13.85	25.72	
ABL	56.17	25.31	79.75	3.38	16.21	28.87	
GL	53.50	17.88	80.98	2.90	17.66	33.0	
Combined	54.34	17.88	80.98	1.77	16.13	29.69	
PL	RV	7.21	6.13	8.45	0.12	0.60	8.33	
ABL	7.15	6.19	8.53	0.10	0.52	7.31	
GL	7.22	6.27	8.72	0.09	0.58	8.15	
Combined	7.21	6.13	8.72	0.06	0.56	7.9	
PD	RV	3.83	2.9	4.82	0.09	0.43	11.33	
ABL	3.90	3.42	4.7	0.06	0.30	7.68	
GL	3.88	3.24	4.82	0.06	0.37	9.68	
Combined	3.87	2.9	4.82	0.04	0.37	9.57	
SPP	RV	9.12	7.61	10.83	0.14	0.68	7.42	
ABL	9.26	8.22	11.0	0.14	0.69	7.54	
GL	9.42	7.67	11.06	0.11	0.70	7.51	
Combined	9.29	7.61	11.06	0.07	0.70	7.53	
SL	RV	4.21	3.47	5.12	0.08	0.40	9.69	
ABL	4.06	3.16	4.99	0.08	0.38	9.55	
GL	4.13	3.23	4.89	0.06	0.39	9.50	
Combined	4.13	3.16	5.12	0.04	0.39	9.55	
SW	RV	3.01	2.29	3.44	0.05	0.26	8.62	
ABL	3.00	2.64	3.71	0.06	0.29	9.97	
GL	3.04	2.57	3.62	0.04	0.26	8.57	
Combined	3.02	2.29	3.71	0.02	0.26	8.89	
HSW	RV	3.09	2.28	4.59	0.10	0.48	15.78	
ABL	2.77	2.03	3.80	0.10	0.50	18.35	
GL	2.86	2.05	3.72	0.07	0.47	16.77	
Combined	2.90	2.03	4.59	0.05	0.50	17.28	
Note:

Water imbibition capacity by pods (WICP), germinated pod % (GP), pre-harvest sprouting (PHS), pod length (PL), pod diameter (PD), number of seeds per pod (SPP), seed length (SL), seed width (SW), 100-seed weight (HSW), released variety (RV), advanced breeding line (ABL), germplasm line (GL).

Genotypic variation for WICP, tolerance to PHS and pod germination

There was a wide variation in WICP, tolerance to PHS and percentage of pods germinated among the assessed genotypes. The water imbibition capacity by pods varied from 32.6% to 68.7% with an average of 46.8% (Table 1). The germinated pods varied between 31% to 95.55% (CV = 15.89%) having an average value of 78.41% (Table 1). Seeds germinated within pod (PHS) ranged between 17.8 to 81% (CV = 29.69%) with a mean value of 54.34%. The highest (>75%) PHS was observed in the genotypes PLM 167, Pusa 0971, KM 7-134, DMS 8, Pusa 1131, M 1358, Satya, KM 22-41 and the lowest (<25%) was observed in the genotypes M 1255, M 145, M 422, M 1421 (all germplasm lines). Among all the genotypes, ten genotypes had PHS of less than 30%. Out of eighty-three genotypes, germplasm line M 1255 was found to be tolerant (<20%) to PHS. Overall, germplasm lines were found to be more tolerant to PHS than the high-yielding released varieties and advanced breeding lines (Fig. 2, Table 2). The variation for tolerance to PHS among genotypes could be further utilized in mungbean improvement program.

Figure 2 Variation in (A) pre-harvest sprouting among released varieties, advanced breeding and germplasm lines (in order of inner to outermost circle) (B) germinated pods (%) among released varieties, advanced breeding and germplasm lines of mungbean genotypes (in order of inner to outermost circle) and (C) PHS tolerant genotype (M 1255) exhibiting less germination in pods, while the susceptible genotype (PLM 167) exhibiting profuse germination.

Table 2 Mungbean genotypes categorized under tolerant (<20%), moderately tolerant (20–40%), moderately susceptible (40–60%) and susceptible (60%) on the basis of seed germination percent in pod (PHS).

Category	Type	Genotype	
Tolerant (<20%)	RV (0)	–	
	ABL (0)	–	
	GL (1)	M 1255	
Moderately tolerant (20–40%)	RV (4)	ML 818, MH 215, SML 668, HUM 2	
	ABL (3)	KM 16-76, KM 16-82, Pusa 1332	
	GL (9)	M 145, M 422, M 1421, M 204, M 981, ML 1451, M 958, OLRM 4, ML 1299	
Moderately susceptible (40–60%)	RV (11)	Pusa Baisakhi, RMG 1028, RMGP 1, Ganga 1, MH 96-1, MH 318, PS 16, IPM 410-3, China Mung, IPM 02-14, RMG 991	
	ABL (10)	MH 1442, IPM 02-15, KM 16-69, MH 934, IPM 02-30, MH 565, Pusa 1132, KM 16-58, Pusa 1342, Pusa 1333	
	GL (9)	M 313, M 1485, IC 436637, M 1400, M 1372, M 684, ML 1464, PLM 271, IC 546476	
Susceptible (>60%)	RV (8)	MH 810, Basanti, PDM 139, COGG 912, Muskan, LGG 460, Satya, Pusa 0971	
	ABL (10)	KM 16-81, Pusa 1341, Pusa 1441, IPM 409-4, TM 9725, ML 1628, KM 22-41, Pusa 1131, DMS 8, KM 7-134	
	GL (18)	ML 2037, IC 546488, M 1156, M 700, M 1168, M 837, IC 436763,
IC 28083, M 703, M 1477, M 1053, M 1370, M 1032, M 1447, M 1131, M 1378, M 1358, PLM 167	
Note:

RV, Released variety, ABL, Advanced breeding lines, GL, Germplasm lines.

Cluster analysis

Cluster analysis was performed for nine quantitative traits to decipher the genetic relationship among the studied genotypes and to identify the desirable genotypes to be utilized for future breeding programs. Cluster analysis grouped the genotypes into seven discrete clusters. Cluster VII was the largest consisting of 53 genotypes (Table 3 and Fig. 3) whereas, Cluster I, being the smallest, consisted of single genotype. Phenotypic variability among the evaluated genotypes was also reflected in the cluster means of studied traits (Table 4). Cluster I comprised of single genotype characterized by the most desirable seed and pod traits in addition to the lowest PHS (17.89) and pod germination (31.11). Cluster II comprised of two genotypes exhibiting maximum WICP (64.2) with short seed length (4.0), seed width (2.77) and low HSW (2.17). Cluster III comprised of eight genotypes characterized by shorter seed length (4.0). Cluster IV comprised of three genotypes characterized by high PHS (67.41%), reduced pod length (6.41) accommodating less number of seeds per pod (7.93). Cluster V also comprised of two genotypes characterized by narrow pod diameter (2.96) and high HSW (4.06). Cluster VI comprised of fourteen genotypes predominantly characterized by longer (7.83) and wider (4.14) pods having longer seeds (4.66). Cluster VII comprised of fifty-three genotypes which exhibited high pod germination (82.47). Genotypic selection for breeding programs should be based on tolerance to PHS and yield-associated traits for a bountiful harvest.

Table 3 Clustering pattern of eighty-three mungbean genotypes.

Cluster	Genotypes	Number of genotypes	
I	M 1255	1	
II	M 422, KM 16-76	2	
III	M 145, M 1421, PUSA 1332, ML 1451, OLRM 4, SML 668, RMG 1028, MH 96-1	8	
IV	SATYA, IC 546488, PUSA 1341	3	
V	PDM 139, BASANTI	2	
VI	RMGP 1, IPM 02-30, M 1372, PLM 271, Pusa Baisakhi, IC 436637, China Mung, IPM 02-14, PUSA 1441, IC 436763, M 703, M 1477, M 1053, ML 1628	14	
VII	M 204, ML 818, M 981, MH 215, M 958, KM 16-82, ML 1299, HUM 2, M 313, MH 1442, IPM 02-15, M 1485, M 1400, KM 16-69, GANGA 1, MH 934, MH 318, PS 16, MH 565, IPM 410-3, PUSA 1132, M 684, ML 1464, KM 16-58, PUSA 1342, PUSA 1333, IC 546476, RMG 991, ML 2037, KM 16-81, MH 810, M 1156, M 700, M 1168, COGG 912, IPM 409-4, M 837, IC 28083, TM 9725, M 1370, MUSKAN, M 1032, LGG 460, M 1447, M 1131, M 1378, KM 2241, M 1358, PUSA 1131, DMS 8, KM 7-134, PUSA 0971, PLM 167	53	

Figure 3 Circular dendrogram.

Circular dendrogram based on pre-harvest sprouting, various seed and pod traits in mungbean genotypes.

Table 4 Cluster means for pre-harvest sprouting, various seed and pod traits in mungbean genotypes.

Cluster	WICP	GP	PHS	PL	PD	SPP	SL	SW	HSW	
I	36.12	31.11	17.89	8.72	4.83	11.06	4.90	3.31	3.41	
II	64.20	45.55	25.00	6.87	3.72	9.22	4.00	2.77	2.17	
III	41.08	63.33	34.60	6.93	4.02	8.79	4.00	2.92	3.03	
IV	58.48	82.22	67.41	6.41	4.12	7.93	4.15	3.30	2.45	
V	54.87	76.67	63.63	6.51	2.96	9.00	4.08	2.87	4.06	
VI	46.37	79.20	57.35	7.83	4.14	9.31	4.66	3.27	3.29	
VII	46.38	82.47	57.23	7.13	3.79	9.43	4.02	2.97	2.78	

Correlation of seed and pod traits with PHS

Correlation analysis revealed that PHS had positive associations with WICP (r = 0.21, p = 0.05), germinated pod % (r = 0.78, p < 0.001), and negative with pod length and seeds per pod (r = −0.13), though not significant (Fig. 4). Pod length exhibited significant positive correlation with pod diameter (r = 0.49, p < 0.001), seed length (r = 0.44, p < 0.001) and number of seeds per pod (r = 0.47, p < 0.001). Seed length and seed width were also positively correlated among (r = 0.60, p < 0.001) and with pod diameter (r = 0.24, p < 0.001; r = 0.33, p < 0.001) (Fig. 4). Genotypes exhibiting <30% PHS (n = 10) were small-seeded having 100-seed weight <3 g (mean HSW- 2.6 g). Bold-seeded genotypes having HSW more than 3.5 g recorded mean PHS to the tune of 57%. However, there was no significant correlation observed between PHS and 100-seed weight.

Figure 4 Correlation among PHS, pod and seed traits.

Water imbibition capacity by pods (WICP), germinated pod % (GP), pre-harvest sprouting (PHS), pod length (PL), pod diameter (PD), number of seeds per pod (SPP), seed length (SL), seed width (SW), 100-seed weight (HSW).

Linear regression model revealed significant positive associations of PHS with water imbibition capacity by pods, germinated pod % and 100-seed weight for the genotypes belonging to group I (<40%). PHS exhibited significant positive associations with germinated pod % and pod diameter whereas negative association with seed width for the genotypes belonging to group III (PHS >60%). Overall, positive association was recorded between PHS and water imbibition capacity by pods, germinated pod % and 100-seed weight (Table 5). Further, principal component analysis (PCA) was carried out to identify the key traits contributing to pre-harvest sprouting in mungbean. The first two principal components for group I (<40%), group II (40–60%), group III (>60%) and overall explained 69.4% and 21.6%, 64.1% and 19.4%, 57.5% and 32.3%, 81.9% and 12% of the total variation for studied traits (Fig. S1). The PCA analysis results confirmed the positive associations of PHS with water imbibition capacity by pods and germinated pod % (Fig. 5). The parameters, pod length, seed length, seed width, pod diameter and 100-seed weight had large positive loadings, whereas WICP had large negative loading on component one. Pre-harvest sprouting and germinated pod % had large positive loadings on component 2 (Fig. 5).

Table 5 Linear regression model explaining associations of pre-harvest sprouting (PHS) with seed and pod traits of mungbean.

Variables	Group I (<40%)	Group II (40–60%)	Group III (>60%)	Overall	
Coefficient	t stat	p value	Coefficient	t stat	p value	Coefficient	t stat	p value	Coefficient	t stat	p value	
Intercept	−34.35	−0.78	0.45	31.04	1.08	0.29	−3.63	−0.10	0.91	−78.31	−2.85	0.005	
WICP	0.61	2.34	0.04	0.14	0.58	0.56	0.21	1.17	0.24	0.72	3.82	0.000	
GP	0.38	3.00	0.01	0.10	0.85	0.40	0.81	3.72	0.000	1.07	12.33	0.000	
PL	2.25	0.44	0.67	−1.16	−0.39	0.69	−1.43	−0.67	0.50	−1.03	−0.36	0.71	
PD	1.92	0.35	0.72	−3.39	−0.88	0.38	5.02	2.02	0.05	5.22	1.46	0.14	
SL	−7.66	−0.62	0.55	−1.74	−0.64	0.52	0.85	0.23	0.81	2.51	0.66	0.50	
SW	5.22	0.24	0.81	5.16	1.13	0.26	−9.51	−2.41	0.02	−4.59	−0.86	0.38	
SPP	−3.23	−1.00	0.34	0.76	0.50	0.62	0.19	0.10	0.91	−1.05	−0.58	0.55	
HSW	12.60	2.85	0.02	3.56	1.16	0.25	2.37	1.38	0.17	5.16	2.11	0.03	
Regression	Multiple R	0.86		Multiple R	0.49		Multiple R	0.74		Multiple R	0.83		
Statistics	Adj. R2	0.46		Adj. R2	−0.03		Adj. R2	0.41		Adj. R2	0.66		
N	17		N	30		N	36		N	83		
p value	0.08		p value	0.55		p value	0.002		p value	<0.001		

Figure 5 Loading plots.

Loading plots (A) group I, (B) group II, (C) group III and (D) all genotypes based on PCA. Water imbibition capacity by pods (WICP), germinated pod % (GP), pre-harvest sprouting (PHS), pod length (PL), pod diameter (PD), number of seeds per pod (SPP), seed length (SL), seed width (SW), 100-seed weight (HSW).

Fresh seed germination (FSG) among selected PHS tolerant and susceptible lines

Genotypes identified as tolerant and susceptible to PHS were subjected to fresh seed germination test. Fresh seed germination among the selected genotypes ranged from 42% (M 204) to 98% (Pusa 1131) (Fig. 6A). Mean FSG among PHS tolerant lines was 54.4% while it was 79% among PHS susceptible lines. Genotypes which exhibited tolerance to PHS recorded relatively higher fresh seed germination which indicates that FSG is largely determined by pod attributes. Results revealed that the PHS tolerant genotypes do not necessarily display deeper fresh seed dormancy per se but the pod wall structure and composition might also be a key factor imparting tolerance to PHS. Inner pod wall of tolerant and susceptible genotypes indicated a clear difference in wax deposition (Fig. 6B) which might be further investigated in detail.

Figure 6 (A) Variation in pre-harvest sprouting (PHS) and fresh seed germination (FSG) among selected PHS tolerant and susceptible lines and (B) the inner pod wall of tolerant and susceptible genotypes showing wax deposition.

Discussion

Breeding for tolerance to pre-harvest sprouting is challenging, owing to polygenic inheritance and discernible genotype and environmental interactions (Li et al., 2021). Pre-harvest sprouting is an environmentally induced phenomenon affected by several factors such as morphological (King & Richards, 1984; Jaiswal et al., 2012; Lin et al., 2016; Rodriguez et al., 2021), biochemical, hormonal, physiological and genetic factors (Tai et al., 2021; Vetch et al., 2019). Identification of genotypes exhibiting tolerance to PHS in addition to higher yield potential is pivotal for minimizing yield losses at the time of harvest and ensuring better end-use quality of the produce. Plethora of literature are available on tolerance to PHS in cereals, however in pulses, particularly mungbean, there are limited studies reported. Hence, the present study was conducted to evaluate the genetic variability for tolerance to pre-harvest sprouting in diverse mungbean genotypes while simultaneously deciphering the association of yield contributing traits with PHS.

Mean performance for released varieties, advanced breeding lines and germplasm lines indicated a wide array of variation among genotypes which can be harnessed by including them in breeding programs. High genetic variability for PHS has been reported in wild Vigna (Lamichaney et al., 2021), urdbean (Lamichaney et al., 2023) and soybean (Kumar et al., 2021). Greater magnitude of genetic variability for PHS has been reported in a similar study by Lamichaney et al. (2018) in mungbean. Out of ten genotypes with least susceptibility to PHS, six were germplasm lines, three were advanced breeding lines while only one released variety ML 818 exhibited less than 30% of PHS. Previous studies have also reported wild accessions/germplasm lines to exhibit higher tolerance to PHS (Nautiyal, Bandyopadhyay & Zala, 2001; Ahmad, Khulbe & Roy, 2014; Sunayana, Ramendra & Raj, 2013; Lamichaney et al., 2018, 2023). The modern mungbean cultivars, developed through long man-made and nature-privileged selection process, have undergone negative selection for hardseededness (physical dormancy) and seed dormancy traits (Laosatit et al., 2022). Negative selection for these traits has made the present-day cultivars yielding high but at the same time vulnerable to various biotic and abiotic stresses including susceptibility to PHS. The variation for PHS in released varieties (range 27–80%; mean 54%) was moderate to high indicating lack of breeding effort for tolerance to PHS in mungbean. Researchers have also documented significant effect of genotype × environment interactions for PHS in wheat (Li et al., 2021). Genotype × environment interactions impose difficulty while phenotyping and breeding for tolerance to PHS. Cluster analysis revealed trait distinctness among eighty-three genotypes. Genotype M 1255 was grouped in a single cluster represented by single genotype exhibiting notable seed and pod traits in addition to low PHS and pod germination. Mohan, Sheeba & Kalaimagal (2021) and Katiyar et al. (2009) underlined the importance of identification of promising genotypes based on high genetic variation for their effective utilization.

In the study, a negative association of WICP with pod length (r = −0.26) and pod diameter (r = −0.33) was observed. This indicated that WICP is determined not by pod surface area but other pod physical characteristics such as presence/ absence of trichomes and trichome density on the pod surface might also play a key role in repelling the water molecules from being imbibed by the pods. Scanning electron microscopic (SEM) study performed by Rao et al. (2023) established the role of ultrastructural features such as presence of trichomes and thick cuticular pod wall in determining resistance to PHS in mungbean.

Positive association between PHS and WICP (r = 0.21) indicated for selecting genotypes which are having pod wall features allowing for minimal water adherence and retention. Further, linear regression model based multivariate analysis revealed positive association between PHS and water imbibition capacity by pods. Though not evaluated intensively in this study, a simple investigation of pod wall indicated differences in wax content on inner pod wall surface of PHS tolerant and susceptible lines. This implies that structure of pod wall might act as a physical barrier restricting water availability to the seeds inside pods and thereby imparting tolerance to PHS in mungbean. Ahmad, Khulbe & Roy (2014) stated high pod wax content and thick pod wall influence water imbibitions. Negative association between pod wall thickness with water imbibition by pod and PHS has been reported previously by Tekrony, Egli & Phillips (1980) and Ahmad, Khulbe & Roy (2014). Rao, Rao & Rao (2007) advocated thick pod wall remains wet for longer compared to thin pod wall providing sufficient moisture to the seeds to germinate within pod and reported significant positive association of pod wall thickness with water imbibition by pods and PHS. Epicuticular wax content of the pod wall and pod wall thickness has been reported to be negatively correlated with PHS in mungbean (Mogali et al., 2023).

For the low (group I, PHS <40%) PHS group, PHS was majorly determined by water imbibition capacity by pods and 100-seed weight whereas for high (group III, PHS >60%) PHS group, pod diameter and seed width were the major determinants to PHS. Thus, not only seed dormancy but other pod and seed traits are also involved in governing PHS in mungbean. Bold-seeded genotypes with 100-seed weight more than 3.5 g were less tolerant to PHS than those of small-seeded genotypes. Multivariate analysis apprised a positive association between PHS and 100-seed weight. Lamichaney et al. (2021, 2023) have also opined that the wild Vigna and urdbean genotypes having higher 100-seed weight to be more susceptible to PHS.

A positive association (r = 0.79) was recorded between fresh seed germination and within pod germination (PHS), though the FSG was relatively higher even for PHS tolerant genotypes. This further affirmed that pod wall morphology and its chemical constituents act as repellent to water molecules thereby conferring partial resistance to PHS. PHS is an interplay between fresh seed germination and seed dormancy. Genotypes having an inherent balance of FSG and seed dormancy would always be desirable to minimize PHS losses. A major QTL on linkage group A, HsA, identified from glasshouse and field data has been reported explaining 23.2% of the dormancy variation in mungbean (Humphry et al., 2005). Four QTLs have been reported for physical seed dormancy in mungbean on four linkage groups (LG1, LG2, LG3 and LG4). QTL Sdwa5.1.1+ (on LG1) exhibited the largest effect explaining 33.7% of the phenotypic variation (Isemura et al., 2012). The allelic contribution for increased seed dormancy at all QTLs was from the cultivated parent (Isemura et al., 2012). Laosatit et al. (2022) identified two linked QTLs, Sdwa5.1.1+ and Sdwa5.1.2+, controlling seed dormancy in wild mungbean. QTL Sdwa5.1.1+ was fine mapped to 3.298-Kb region containing gene LOC106767068, designated as VrKNAT7-1 which controlled physical seed dormancy.

Conclusions

The genotypes investigated in this study exhibited wide genetic variability for tolerance to PHS and yield associated traits which can be harnessed in mungbean breeding programs for improving yield, seed viability and tolerance to PHS. Genotypes M 1255, M 145, M 422 and M 1421 were found promising and can be availed as mungbean genetic resources/donors for incorporating tolerance to PHS. The identified lines may be employed in development of mapping populations to map the genes/QTLs and decipher candidate genes for PHS in mungbean. These germplasm lines can be hybridized with elite lines to broaden their genetic base. Significant correlation of PHS tolerance with yield associated traits may aid for simultaneously improving for both the traits. Pod wall structure may be further investigated to get insights on ultrastructural features determining tolerance to PHS. Genotypes tolerant and susceptible to PHS can be investigated to gain insight into biochemical pathways, hormonal regulation and molecular mechanisms involved.

Supplemental Information

Supplemental Information 1 Raw data of traits studied (mean values) and selected genotypes list.

Supplemental Information 2 Scree plot A) group I, B) group II, C) group III and D) overall.

Supplemental Information 3 List of 83 mungbean genotypes along with their source.

The authors are grateful for the facility needed for the smooth conduct of research to the ICAR-IARI, New Delhi.

Additional Information and Declarations

Competing Interests

Author Contributions

Data Availability

The authors declare that they have no competing interests.

Soma Gupta conceived and designed the experiments, performed the experiments, analyzed the data, prepared figures and/or tables, authored or reviewed drafts of the article, and approved the final draft.

Muraleedhar Aski performed the experiments, analyzed the data, prepared figures and/or tables, and approved the final draft.

Gyan Prakash Mishra conceived and designed the experiments, authored or reviewed drafts of the article, and approved the final draft.

Prachi S. Yadav conceived and designed the experiments, performed the experiments, prepared figures and/or tables, authored or reviewed drafts of the article, and approved the final draft.

Kuldeep Tripathi performed the experiments, prepared figures and/or tables, and approved the final draft.

Sandeep Kumar Lal performed the experiments, prepared figures and/or tables, and approved the final draft.

Simran Jain performed the experiments, analyzed the data, prepared figures and/or tables, and approved the final draft.

Ramakrishnan Madhavan Nair conceived and designed the experiments, authored or reviewed drafts of the article, and approved the final draft.

Harsh Kumar Dikshit conceived and designed the experiments, authored or reviewed drafts of the article, and approved the final draft.

The following information was supplied regarding data availability:

The raw data is available in the Supplemental Files.

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
