# Peer review of "Genetic variation for tolerance to pre-harvest sprouting in mungbean (Vigna radiata) genotypes"

_PeerJ, doi:10.7717/peerj.17609_

## Round 0.1 · original submission · Major Revisions

We have received comments. The current version can not be accepted. Revise the article considering the reviewers' comments. The main concerns are the lack of replication in the experiment and weak language.

**Language Note:** The Academic Editor has identified that the English language must be improved. PeerJ can provide language editing services - please contact us at [email protected] for pricing (be sure to provide your manuscript number and title). Alternatively, you should make your own arrangements to improve the language quality and provide details in your response letter. – PeerJ Staff

Reviewer 1 ·

Basic reporting

no comment

Experimental design

Please, Use new analyses and tests. A similar article was published in 2017. So this article should be very interesting.

Validity of the findings

no comment

Additional comments

Use new analyses and tests. A similar article was published in 2017. So this article should be very interesting.

Annotated reviews are not available for download in order to protect the identity of reviewers who chose to remain anonymous.

Reviewer 2 ·

Basic reporting

The manuscript gave a very good explanation of the genetic variability for tolerance to pre-harvest sprouting, which is a serious threat, in diverse mungbean genotypes and the association of yield-contributing traits with pre-harvest sprouting, which makes it more interesting. First and foremost, I commend you on the clarity and organization of your writing. Your manuscript is well-structured, making it easy for readers to follow the flow of ideas. The logical progression from introduction to conclusion demonstrates a thoughtful and cohesive approach to presenting your research. I also want to acknowledge the clarity of your figures and tables, which greatly enhance the readability of your manuscript. Your methodology is robust, and the data analysis appears comprehensive. The thoroughness of your research design strengthens the validity of your findings, contributing significantly to the overall quality of the manuscript. The visual representations effectively complement the textual content, providing a well-rounded presentation of your research. However, I would like to kindly highlight a few minor revisions for improvement and offer constructive suggestions to enhance the overall quality of the manuscript. They are listed below:
1. There are a few spacing errors in some places in the manuscript.
2. Check if the values mentioned in lines 162-165 are consistent with the values in Table 1? You may try to highlight the value. It may confuse the reader.

Experimental design

No comment

Validity of the findings

Valid

Additional comments

No comment

Reviewer 3 ·

Basic reporting

The Authors explored the genetic variability for PHS in mungbean in a collection of 83 cultivated materials. This objective is agronomically important as mungbean is a highly important crop worldwide, and the identification of resistant materials can help breeding to improve PHS resistance in mungbean. As the Authors mention in the introduction, PHS is a complex trait regulated by many genes and strongly affected by the environment. The identification of resistant and susceptible varieties and lines among released and breeding lines might be useful for future mapping and breeding. In addition to studying PHS in 83 diverse mungbean genotypes, the Authors explore relationships between PHS with several seed and pod traits (seed size, hardseededness, pod size and morphology, water imbibed by pods). Regarding the investigated associations between PHS and seed / pod characteristics, the results reported here appear to confirm those from previous works (cited in the discussion section) supporting that different resistance to PHS among cultivated mungbeans is more dependent on pod characteristics affecting imbibition and water retention rather than variability in seed hardseededness.

Experimental design

Because PHS is a quantitative trait with significant environmental effect it is surprising that a single field experiment was conducted with no field repetitions at all (a single plot-row was performed for each genotype). Data presented for the whole collection belongs to one field trial only. This is partially compensated by a second trial where only a selection of materials that exhibited extreme PHS phenotypes in the first trial were cultivated. In this second trial, fresh seed germination was tested to confirm that this trait is not related to PHS resistance in mungbean.
To provide a robust characterization of materials regarding PHS behavior, field replications are necessary. Otherwise, lack of replications should be justified properly as part of an exploratory (preliminary) work.

Validity of the findings

Because of the experimental design (no field repetitions and only one environment), the results can be considered as preliminary to recommend any of the tested materials for breeding purposes.

Additional comments

The introduction can be greatly improved if the Authors clarify better what does this study add or confirm as compared to previous published works in mungbean. A comprehensive and very detailed study was performed by Ahmad, Khulbe & Roy (2014) with similar objectives, but is not mentioned properly in the introduction section (although it is mentioned later in the discussion).
The introduction also reflects a lack of clarity regarding the conceptual framework on dormancy in general but also in mungbean. For example, a clear definition of physical dormancy is not provided here, while hormonal regulation of seed dormancy is presented with more than enough detail despite it is not relevant in the context of physical dormancy.
In the discussion section, concepts regarding dormancy are again confusing. Please define clearly physical dormancy, and distinguish from physiological dormancy which is regulated hormonally (and is not the case in mungbean). Discussion about dormancy QTL in cereals is not really helpful as it is. I suggest to focus on comparison of your results and previous works in mungbean, and what is known about the main factors that contribute to PHS in this species and within the collection of genotypes used in this study.

Some specific comments:
L.86-92. The Authors mention literature on other species where QTL for PHS have been mapped, but when referring to mungbean they only say that literature is limited and don´t mention an important and recent work by Laosatit et al. (2022). This work is indeed cited in the discussion section but only to affirm that mungbean has undergone negative selection for hardseededness.
L42. Need to define HSW here.
L70-73. Please indicate if you are referring to PHS in mungbean in particular and provide proper citations for mungbean. If not, please provide citations for more foundational works or comprehensive reviews on the topic. This sentence is a bit confusing as it refers to hard seededness and hormonal balance as different traits from seed dormancy. Useful definitions for seed dormancy were provided by these authors:
Baskin & Baskin 2007 (classification system for seed dormancy)
Finch-Savage, W. E., & Leubner-Metzger, G. (2006). Seed dormancy and the control of germination. The New phytologist, 171(3), 501–523. https://doi.org/10.1111/j.1469-
8137.2006.01787.x.
Hilhorst, H.W.M.; Karssen, C.M. Seed dormancy and germination: The role of abscisic acid and gibberellins and the importance of hormone mutants. Plant Growth Regul. 1992, 11, 225– 238.
L 84-85. The citations provided here do not support that ethylene and BR modulate PHS. I think that hormonal regulation of seed dormancy is not really important in the context of physical dormancy as is the case in mung bean, and their mention here is not justified.
L 121. Percentage of fresh weight or dry weight? Also, remove "Percent seed germination..." as this belongs to the section below (Line 123)
L167. Need to define WIC at first mention in the text. This and all traits measured should be identified in the materials and methods section.
L 182-183. “…and negative with pod length and seeds per pod (r= - 0.13)…”. Was this significant? Please include p values or identify significance of correlations in Table 2. Only mention significant correlations in the text or indicate clearly if not significant.
Figure 4: This figure may be as Supplementary Data.
L 213-214. instead of "lack fresh seed dormancy.." I think you meant to say "...display deeper fresh seed dormancy per se but...". Please revise.

---

## Round 0.2 · accepted · Accept

The Reviewers have given accept decision for your revised article.

Reviewer 1 ·

Basic reporting

I have no further suggestions to make, and believe that the authors have addressed all of my comments and that the manuscript is now ready to publish.

Experimental design

I have no further suggestions to make, and believe that the authors have addressed all of my comments and that the manuscript is now ready to publish.

Validity of the findings

I have no further suggestions to make, and believe that the authors have addressed all of my comments and that the manuscript is now ready to publish.

Additional comments

I have no further suggestions to make, and believe that the authors have addressed all of my comments and that the manuscript is now ready to publish.

Reviewer 2 ·

Basic reporting

All the chages suggested have been incorporated in the revised manuscript.

Experimental design

no comment

Validity of the findings

no comment

Additional comments

no comment